

# Gender differences and bias in open source: pull request acceptance of women versus men

Josh Terrell[1], Andrew Kofink[2], Justin Middleton[2], Clarissa Rainear[2], Emerson Murphy-Hill[2], Chris Parnin[2] and Jon Stallings[3]

[1] Department of Computer Science, California Polytechnic State University—San Luis Obispo, San Luis Obispo, CA, United States
[2] Department of Computer Science, North Carolina State University, Raleigh, NC, United States
[3] Department of Statistics, North Carolina State University, Raleigh, NC, United States

## ABSTRACT

Biases against women in the workplace have been documented in a variety of studies. This paper presents a large scale study on gender bias, where we compare acceptance rates of contributions from men versus women in an open source software community. Surprisingly, our results show that women's contributions tend to be accepted more often than men's. However, for contributors who are outsiders to a project and their gender is identifiable, men's acceptance rates are higher. Our results suggest that although women on GitHub may be more competent overall, bias against them exists nonetheless.

## INTRODUCTION

In 2012, a software developer named Rachel Nabors wrote about her experiences trying to fix bugs in open source software (http://rachelnabors.com/2012/04/of-github-and-pull-requests-and-comics/). Nabors was surprised that all of her contributions were rejected by the project owners. A reader suggested that she was being discriminated against because of her gender.

Research suggests that, indeed, gender bias pervades open source. In Nafus' interviews with women in open source, she found that "sexist behavior is…as constant as it is extreme" (*Nafus, 2012*). In Vasilescu and colleagues' study of Stack Overflow, a question and answer community for programmers, they found "a relatively 'unhealthy' community where women disengage sooner, although their activity levels are comparable to men's" (*Vasilescu, Capiluppi & Serebrenik, 2014*). These studies are especially troubling in light of recent research which suggests that diverse software development teams are more productive than homogeneous teams (*Vasilescu et al., 2015*). Nonetheless, in a 2013 survey of the more than 2000 open source developers who indicated a gender, only 11.2% were women (*Arjona-Reina, Robles & Dueas, 2014*).

Corresponding author
Emerson Murphy-Hill,
emerson@csc.ncsu.edu

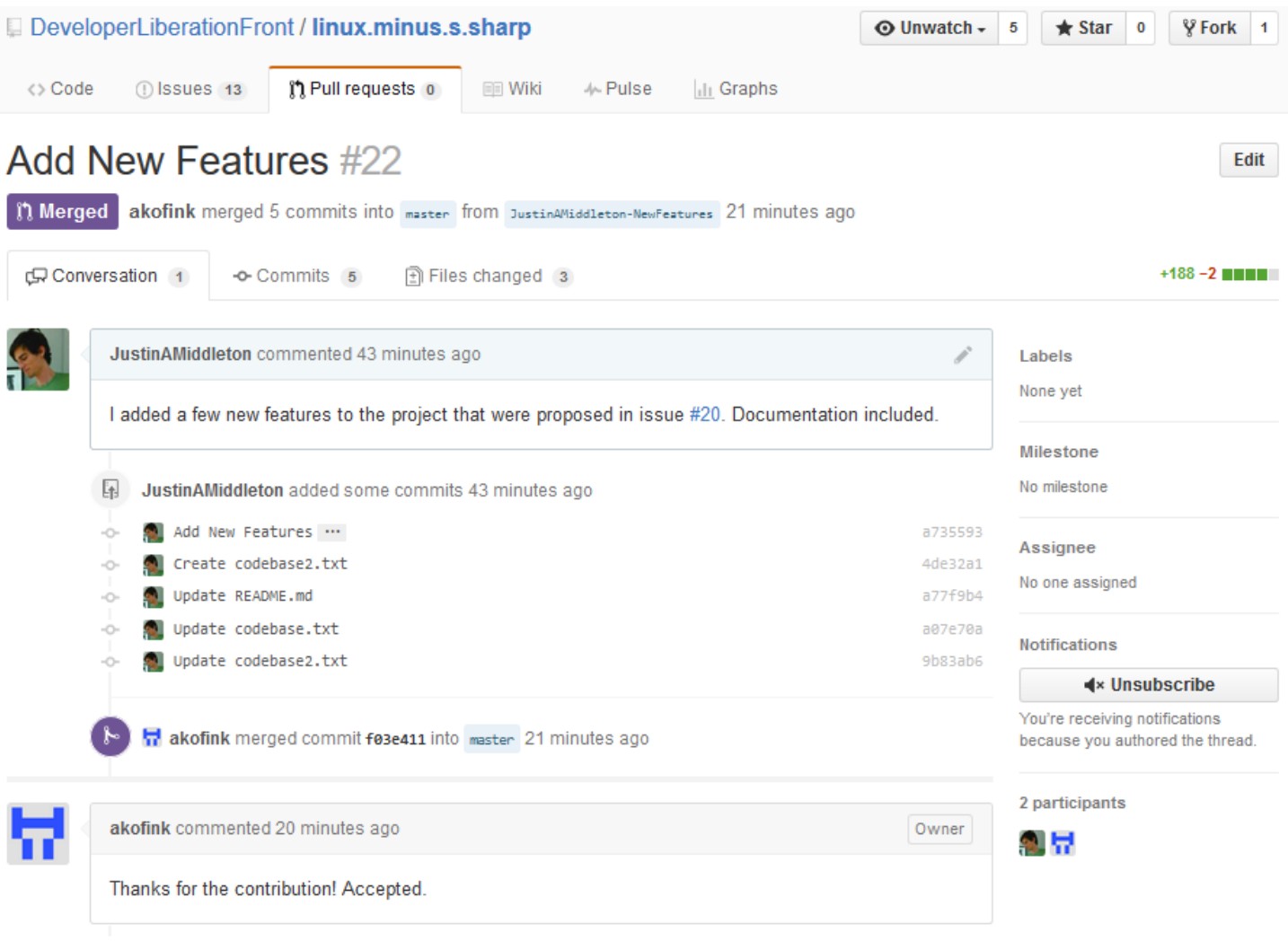

**Figure 1** **GitHub user 'JustinAMiddleton' makes a pull request; the repository owner 'akofink' accepts it by merging it.** The changes proposed by JustinAMiddleton are now incorporated into the project.

This article presents an investigation of gender bias in open source by studying how software developers respond to *pull requests*, proposed changes to a software project's code, documentation, or other resources. A successfully accepted, or 'merged,' example is shown in Fig. 1. We investigate whether pull requests are accepted at different rates for self-identified women compared to self-identified men. For brevity, we will call these developers 'women' and 'men,' respectively. Our methodology is to analyze historical GitHub data to evaluate whether pull requests from women are accepted less often. While other open source communities exist, we chose to study GitHub because it is the largest (*Gousios et al., 2014*), claiming to have over 12 million collaborators across 31 million software repositories (https://github.com/about/press).

The main contribution of this paper is an examination of gender differences and bias in the open source software community, enabled by a novel gender linking technique that

associates more than 1.4 million community members to self-reported genders. To our knowledge, this is the largest scale study of gender bias to date in open source communities.

## RELATED WORK

A substantial part of activity on GitHub is done in a professional context, so studies of gender bias in the workplace are relevant. Because we cannot summarize all such studies here, we instead turn to Davison and Burke's meta-analysis of 53 papers, each studying between 43 and 523 participants, finding that male and female job applicants generally received lower ratings for opposite-sex-type jobs (e.g., nurse is a female sex-typed job, whereas carpenter is male sex-typed) (*Davison & Burke, 2000*).

The research described in Davison and Burke's meta-analysis can be divided into experiments and field studies. Experiments attempt to isolate the effect of gender bias by controlling for extrinsic factors, such as level of education. For example, *Knobloch-Westerwick, Glynn & Huge (2013)* asked 243 scholars to read and evaluate research paper abstracts, then systematically varied the gender of each author; overall, scholars rated papers with male authors as having higher scientific quality. In contrast to experiments, field studies examine existing data to infer where gender bias may have occurred retrospectively. For example, Roth and colleagues' meta-analysis of such studies, encompassing 45,733 participants, found that while women tend to receive better job performance ratings than men, women also tend to be passed up for promotion (*Roth, Purvis & Bobko, 2012*).

Experiments and retrospective field studies each have advantages. The advantage of experiments is that they can more confidently infer cause and effect by isolating gender as the predictor variable. The advantage of retrospective field studies is that they tend to have higher ecological validity because they are conducted in real-world situations. In this paper, we use a retrospective field study as a first step to quantify the effect of gender bias in open source.

Several other studies have investigated gender in the context of software development. Burnett and colleagues (*2010*) analyzed gender differences in 5 studies that surveyed or interviewed a total of 2,991 programmers; they found substantial differences in software feature usage, tinkering with and exploring features, and in self-efficacy. *Arun & Arun (2002)* surveyed 110 Indian software developers about their attitudes to understand gender roles and relations but did not investigate bias. Drawing on survey data, Graham and Smith demonstrated that women in computer and math occupations generally earn only about 88% of what men earn (*Graham & Smith, 2005*). Lagesen contrasts the cases of Western versus Malaysian enrollment in computer science classes, finding that differing rates of participation across genders results from opposing perspectives of whether computing is a "masculine" profession (*Lagesen, 2008*). The present paper builds on this prior work by looking at a larger population of developers in the context of open source communities.

Some research has focused on differences in gender contribution in other kinds of virtual collaborative environments, particularly Wikipedia. Antin and colleagues (*2011*). followed the activity of 437 contributors with self-identified genders on Wikipedia and found that, of the most active users, men made more frequent contributions while women made larger contributions.

There are two gender studies about open source software development specifically. The first study is Nafus' anthropological mixed-methods study of open source contributors, which found that "men monopolize code authorship and simultaneously de-legitimize the kinds of social ties necessary to build mechanisms for women's inclusion", meaning values such as politeness are favored less by men (*Nafus, 2012*). The other is Vasilescu and colleagues' *(2015)* study of 4,500 GitHub contributors, where they inferred the contributors' gender based on their names and locations (and validated 816 of those genders through a survey); they found that gender diversity is a significant and positive predictor of productivity. Our work builds on this by investigating bias systematically and at a larger scale.

## GENERAL METHODOLOGY

Our main research question was

> To what extent does gender bias exist when pull requests are judged on GitHub?

We answer this question from the perspective of a *retrospective cohort study*, a study of the differences between two groups previously exposed to a common factor to determine its influence on an outcome (*Doll, 2001*). One example of a similar retrospective cohort study was Krumholz and colleagues' *(1992)*. review of 2,473 medical records to determine whether there exists gender bias in the treatment of men and women for heart attacks. Other examples include the analysis of 6,244 school discipline files to evaluate whether gender bias exists in the administration of corporal punishment (*Gilbert, Williams & Lundberg, 1994*) and the analysis of 1,851 research articles to evaluate whether gender bias exists in the peer reviewing process for the Journal of the American Medical Association (*Shaw & Braden, 1990*).

To answer the research question, we examined whether men and women are equally likely to have their pull requests accepted on GitHub, then investigated why differences might exist. While the data analysis techniques we used were specific to each approach, there were several commonalities in the data sets that we used, as we briefly explain below. For the sake of maximizing readability of this paper, we describe our methodology in detail in the 'Material and Methods' Appendix.

We started with a GHTorrent (*Gousios, 2013*) dataset that contained public data on pull requests from June 7, 2010 to April 1, 2015, as well as data about users and projects. We then augmented this GHTorrent data by mining GitHub's webpages for information about each pull request status, description, and comments.

GitHub does not request information about users' genders. While previous approaches have used gender inference (*Vasilescu, Capiluppi & Serebrenik, 2014*; *Vasilescu et al., 2015*), we took a different approach—linking GitHub accounts with social media profiles where the user has self-reported gender. Specifically, we extract users' email addresses from GHTorrent, look up that email address on the Google+ social network, then, if that user has a profile, extract gender information from these users' profiles. Out of 4,037,953

GitHub user profiles with email addresses, we were able to identify 1,426,127 (35.3%) of them as men or women through their public Google+ profiles. We are the first to use this technique, to our knowledge.

We recognize that our gender linking approach raises privacy concerns, which we have taken several steps to address. First, this research has undergone human subjects IRB review, research that is based entirely on publicly available data. Second, we have informed Google about our approach in order to determine whether they believe our approach to linking email addresses to gender is a privacy violation of their users; they responded that it is consistent with Google's terms of service (https://sites.google.com/site/bughunteruniversity/nonvuln/discover-your-name-based-on-e-mail-address). Third, to protect the identities of the people described in this study to the extent possible, we do not plan to release our data that links GitHub users to genders.

## RESULTS

We describe our results in this section; data is available in Supplemental Files.

### Are women's pull requests less likely to be accepted?

We hypothesized that pull requests made by women are less likely to be accepted than those made by men. Prior work on gender bias in hiring—that a job application with a woman's name is evaluated less favorably than the same application with a man's name (*Moss-Racusin et al., 2012*)—suggests that this hypothesis may be true.

To evaluate this hypothesis, we looked at the pull status of every pull request submitted by women compared to those submitted by men. We then calculate the merge rate and corresponding confidence interval, using the Clopper–Pearson exact method (*Clopper & Pearson, 1934*), and find the following:

| Gender | Open | Closed | Merged | Merge Rate | 95% Confidence interval |
|--------|------|--------|--------|------------|-------------------------|
| Women | 8,216 | 21,890 | 111,011 | **78.7%** | [78.45%,78.88%] |
| Men | 150,248 | 591,785 | 2,181,517 | **74.6%** | [74.57%,74.67%] |

The hypothesis is not only false, but it is in the opposite direction than expected; *women tend to have their pull requests accepted at a higher rate than men*! This difference is statistically significant ($\chi^2(df = 1, n = 3,064,667) = 1,170, p < .001$). What could explain this unexpected result?

### *Open source effects*

Perhaps our GitHub data are not representative of the open source community; while all projects we analyzed were public, not all of them are licensed as open source. Nonetheless, if we restrict our analysis to just projects that are explicitly licensed as open source, women continue to have a higher acceptance rate ($\chi^2(df = 1, n = 1,424,127) = 347, p < .001$):

| Gender | Open | Closed | Merged | **Merge Rate** | 95% Confidence Interval |
|--------|------|--------|--------|-----------|-------------------------|
| Women | 1,573 | 7,669 | 32,944 | **78.1%** | [77.69%,78.49%] |
| Men | 60,476 | 297,968 | 1,023,497 | **74.1%** | [73.99%,74.14%] |

### Insider effects

Perhaps women's high acceptance rate is because they are already well known in the projects they make pull requests in. Pull requests can be made by anyone, including both insiders (explicitly authorized owners and collaborators) and outsiders (other GitHub users). If we exclude insiders from our analysis, the women's acceptance rate (62.1% [61.65%,62.53%]) continues to be significantly higher than men's (60.7% [60.65%,60.82%]) ($\chi^2(df = 1, n = 1,372,834) = 35, p < .001$).

### Experience effects

Perhaps only a few highly successful and prolific women, responsible for a substantial part of overall success, are skewing the results. To test this, we calculated the pull request acceptance rate for each woman and man with 5 or more pull requests, then found the average acceptance rate across those two groups. The results are displayed in Fig. 2. We notice that women tend to have a bimodal distribution, typically being either very successful ($>$90% acceptance rate) or unsuccessful ($<$10%). But these data tell the same story as the overall acceptance rate; women are more likely than men to have their pull requests accepted.

Why might women have a higher acceptance rate than men, given the gender bias documented in the literature? In the remainder of this section, we will explore this question by evaluating several hypotheses that might explain the result.

## Do women's pull request acceptance rates start low and increase over time?

One plausible explanation is that women's first few pull requests get rejected at a disproportionate rate compared to men's, so they feel dejected and do not make future pull requests. This explanation is supported by Reagle's account of women's participation in virtual collaborative environments, where an aggressive argument style is necessary to justify one's own contributions, a style that many women may find to be not worthwhile (*Reagle, 2012*). Thus, the overall higher acceptance rate for women would be due to survivorship bias within GitHub; the women who remain and do the majority of pull requests would be better equipped to contribute, and defend their contributions, than men. Thus, we might expect that women have a lower acceptance rate than men for early pull requests but have an equivalent acceptance rate later.

To evaluate this hypothesis, we examine pull request acceptance rate over time, that is, the mean acceptance rate for developers on their first pull request, second pull request, and so on. Figure 3 displays the results. Orange points represent the mean acceptance rate for women, and purple points represent acceptance rates for men. Shaded regions indicate the pointwise 95% Clopper–Pearson confidence interval.

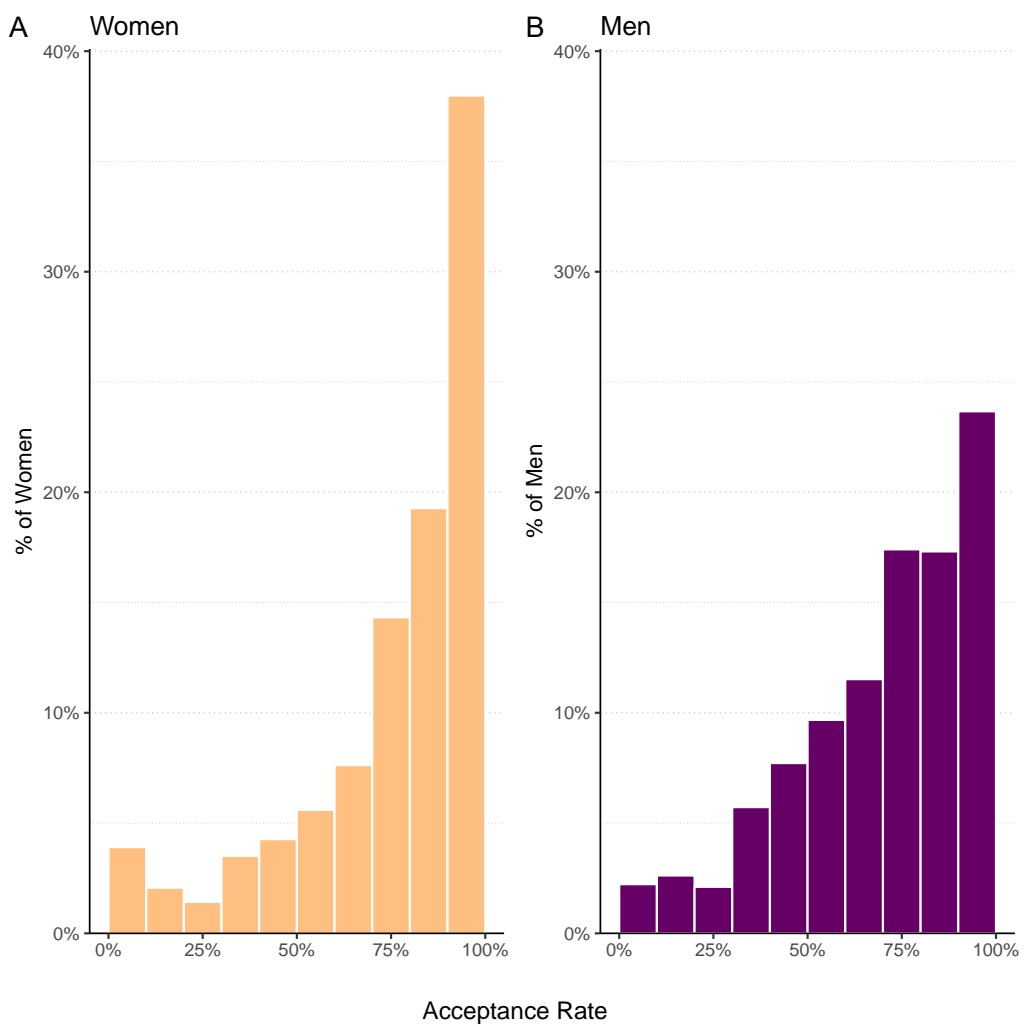

**Figure 2** Histogram of mean acceptance rate per developer for women (mean 76.9%, median 84.9%) and men (mean 71.0%, median 76.0%).

While developers making their initial pull requests do get rejected more often, women generally still maintain a higher rate of acceptance throughout. The acceptance rate of women tends to fluctuate at the right of the graph, because the acceptance rate is affected by only a few individuals. For instance, at 128 pull requests, only 103 women are represented. Intuitively, where the shaded region for women includes the corresponding data point for men, the reader can consider the data too sparse to conclude that a substantial difference exists between acceptance rates for women and men. Nonetheless, between 1 and 64 pull requests, women's higher acceptance rate remains. Thus, the evidence casts doubt on our hypothesis.

### Are women focusing their efforts on fewer projects?

One possible explanation for women's higher acceptance rates is that they are focusing their efforts more than men; perhaps their success is explained by doing pull requests on few projects, whereas men tend to do pull requests on more projects. First, the data do

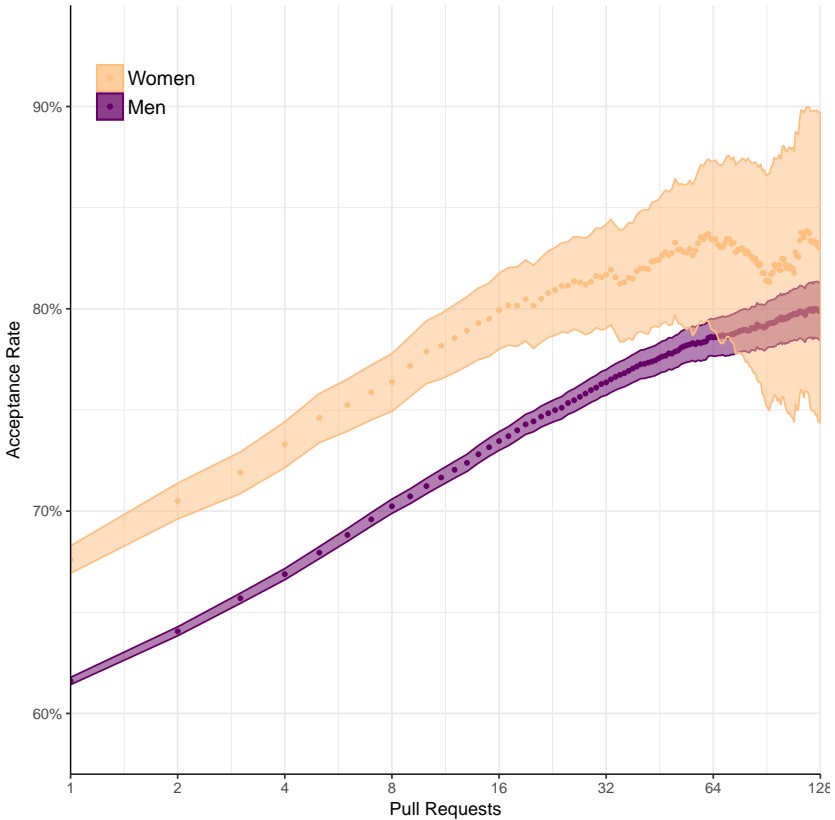

**Figure 3** Pull request acceptance rate over time.

suggest that women tend to contribute to fewer projects than men. While the median number of projects contributed to via pull request is 1 for both genders (that is, the 50th percentile of developers); at the 75th percentile it is 2 for women and 3 for men, and at the 90th percentile it is 4 for women and 7 for men.

But the fact that women tend to contribute to fewer projects does not explain why women tend to have a higher acceptance rate. To see why, consider Fig. 4; on the *y* axis is mean acceptance rate by gender, and on the *x* axis is number of projects contributed to. When contributing to between 1 and 5 projects, women have a higher acceptance rate as they contribute to more projects. Beyond 5 projects, the 95% confidence interval indicates women's data are too sparse to draw conclusions confidently.

## Are women making pull requests that are more needed?

Another explanation for women's pull request acceptance rate is that, perhaps, women disproportionately make contributions that projects need more specifically. What makes a contribution "needed" is difficult to assess from a third-party perspective. One way is to look at which pull requests link to issues in projects' GitHub issue trackers. If a pull request references an issue, we consider it to serve a more specific and recognized need than an otherwise comparable one that does not. To support this argument with data, we randomly selected 30 pull request descriptions that referenced issues; in 28 cases, the reference was

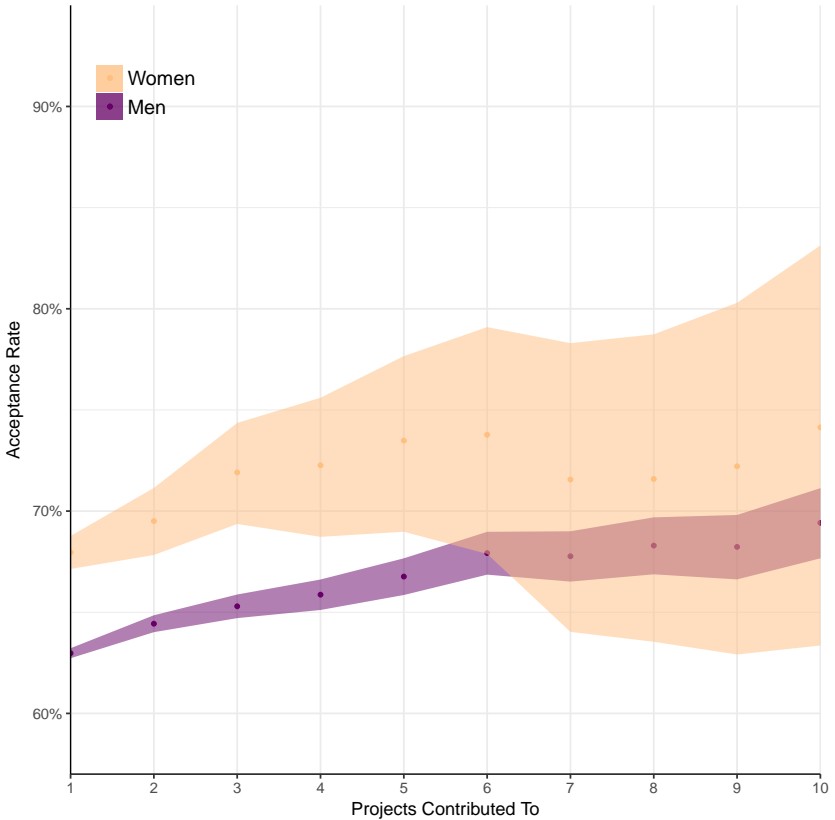

**Figure 4** **Pull request acceptance rate by number of projects contributed to.**

an attempt to fix all or part of an issue. Based on this high probability, we can assume that when someone references an issue in a pull request description, they usually intend to fix a specific problem in the project. Thus, if women more often submit pull requests that address an documented need and this is enough to improve acceptance rates, we would expect that these same requests are more often linked to issues.

We evaluate this hypothesis by parsing pull request descriptions and calculating the percentage of pulls that reference an issue. To eliminate projects that do not use issues or do not customarily link to them in pull requests, we analyze only pull requests in projects that have at least one linked pull request. Here are the results:

| Gender | Without reference | With reference | % | 95% Confidence Interval |
|---|---|---|---|---|
| Women | 33,697 | 4,748 | 12.4% | [12.02%,12.68%] |
| Men | 1,196,519 | 182,040 | 13.2% | [13.15%,13.26%] |

This data show a statistically significant difference ($\chi^2(df = 1, n = 1,417,004) = 24$, $p < .001$). Contrary to the hypothesis, women are slightly less likely to submit a pull request that mentions an issue, suggesting that women's pull requests are less likely to fulfill an documented need. Note that this does not imply women's pull requests are less valuable,

but instead that the need they fulfill appears less likely to be recognized and documented before the pull request was created. Regardless, the result suggests that women's increased success rate is not explained by making more specifically needed pull requests.

### Are women making smaller changes?

Maybe women are disproportionately making small changes that are accepted at a higher rate because the changes are easier for project owners to evaluate. This is supported by prior work on pull requests suggesting that smaller changes tend to be accepted more than larger ones (*Gousios, Pinzger & Deursen, 2014*).

We evaluated the size of the contributions by analyzing lines of code, modified files, and number of commits included. The following table lists the median and mean lines of code added, removed, files changed, and commits across 3,062,677 pull requests:

|        |           | Lines added    | Lines removed | Files changed | Commits    |
|--------|-----------|----------------|---------------|---------------|------------|
| Women  | Median    | 29             | 5             | 2             | 1          |
|        | Mean      | 1,591          | 597           | 29.2          | 5.2        |
| Men    | Median    | 20             | 4             | 2             | 1          |
|        | Mean      | 1,003          | 431           | 26.8          | 4.8        |
| $t$-test | Statistic | 5.74         | 3.03          | 1.52          | 7.36       |
|        | df        | 146,897        | 149,446       | 186,011       | 155,643    |
|        | p         | < .001         | 0.0024554     | 0.12727       | < .001     |
|        | CI        | [387.3,789.3]  | [58.3,272]    | [−0.7,5.4]    | [0.3,0.5]  |

The bottom of this chart includes Welch's $t$-test statistics, comparing women's and men's metrics, including 95% confidence intervals for the mean difference. For three of four measures of size, women's pull requests are significantly larger than men's.

One threat to this analysis is that lines added or removed may exaggerate the size of a change whenever a refactoring is performed. For instance, if a developer moves a 1,000-line class from one folder to another, even though the change may be relatively benign, the change will show up as 1,000 lines added and 1,000 lines removed. Although this threat is difficult to mitigate definitively, we can begin to address it by calculating the net change for each pull request as the number of added lines minus the number of removed lines. Here is the result:

|         |           | Net lines changed |
|---------|-----------|-------------------|
| Women   | Median    | 11                |
|         | Mean      | 995               |
| Men     | Median    | 7                 |
|         | Mean      | 571               |
| $t$-test | Statistic | 4.06             |
|         | df        | 148,010           |
|         | p         | < .001            |
|         | CI        | [218.9,627.4]     |

This difference is also statistically significant. So even in the face of refactoring, the conclusion holds: women make pull requests that add and remove more lines of code, and contain more commits. This is consistent with larger changes women make on Wikipedia (*Antin et al., 2011*).

### Are women's pull requests more successful when contributing code?

One potential explanation for why women get their pull requests accepted more often is that the *kinds* of changes they make are different. For instance, changes to HTML could be more likely to be accepted than changes to C code, and if women are more likely to change HTML, this may explain our results. Thus, if we look only at acceptance rates of pull requests that make changes to program code, women's high acceptance rates might disappear. For this, we define program code as files that have an extension that corresponds to a Turing-complete programming language. We categorize pull requests as belonging to a single type of source code change when the majority of lines modified were to a corresponding file type. For example, if a pull request changes 10 lines in `.js` (javascript) files and 5 lines in `.html` files, we include that pull request and classify it as a `.js` change. Figure 5 shows the results for the 10 most common programming language files (Fig. 5A) and the 10 most common non-programming language files (Fig. 5B). Each pair of bars summarizes pull requests classified as part of a programming language file extension, where the height of each bar represents the acceptance rate and each bar contains a 95% Clopper–Pearson confidence interval. An asterisk (*) next to a language indicates a statistically significant difference between men and women for that language using a chi-squared test, after a Benjamini–Hochberg correction (*Benjamini & Hochberg, 1995*) to control for false discovery.

Overall, we observe that women's acceptance rates are higher than men's for almost every programming language. The one exception is .m, which indicates Objective-C and Matlab, for which the difference is not statistically significant.

### Is a woman's pull request accepted more often because she appears to be a woman?

Another explanation as to why women's pull requests are accepted at a higher rate would be what McLoughlin calls Type III bias: "the singling out of women by gender with the intention to help" (*McLoughlin, 2005*). In our context, project owners may be biased towards wanting to help women who submit pull requests, especially outsiders to the project. In contrast, male outsiders without this benefit may actually experience the opposite effect, as distrust and bias can be stronger in stranger-to-stranger interactions (*Landy, 2008*). Thus, we expect that women who can be perceived as women are more likely to have their pull requests accepted than women whose gender cannot be easily inferred, especially when compared to male outsiders.

We evaluate this hypothesis by comparing pull request acceptance rate of developers who have gender-neutral GitHub profiles and those who have gendered GitHub profiles. We define a gender-neutral profile as one where a gender cannot be readily inferred from their profile. Figure 1 gives an example of a gender-neutral GitHub user, "akofink", who

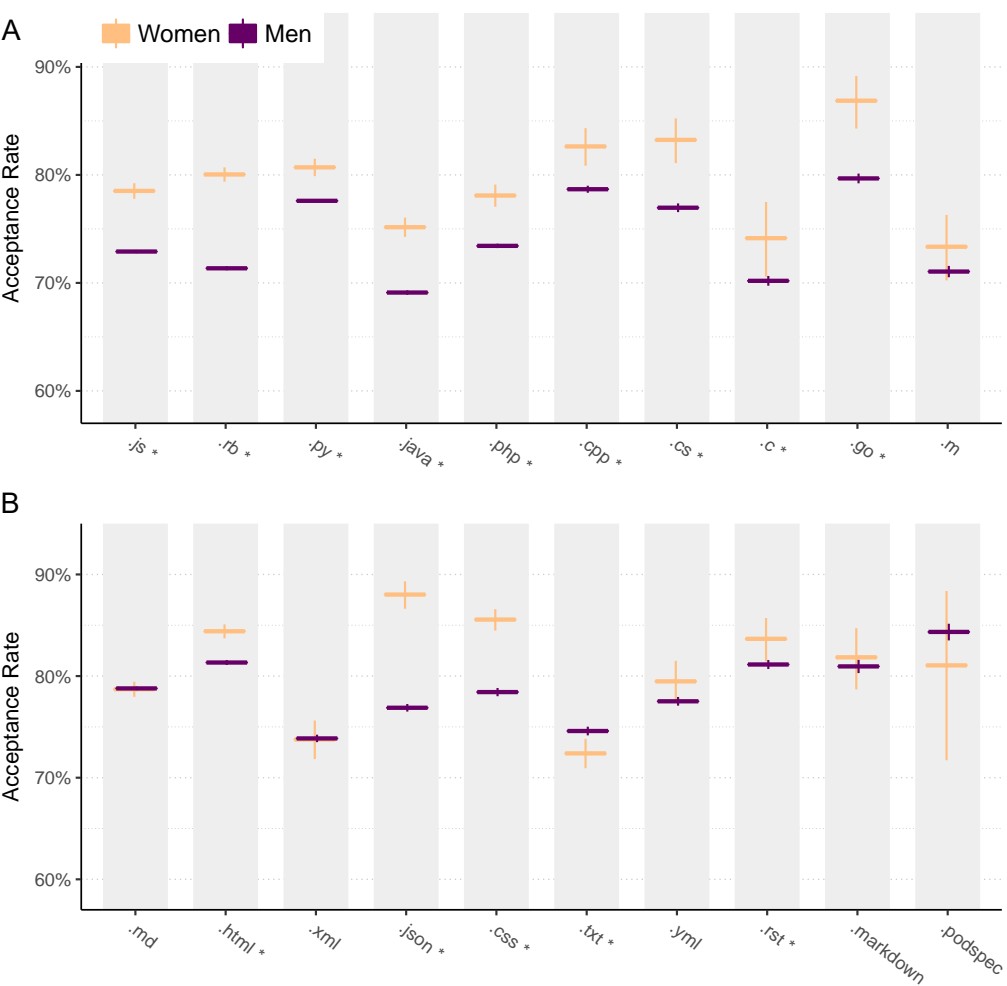

**Figure 5  Pull request acceptance rate by file type, for programming languages (A) and non-programming languages (B).**

uses an *identicon*, an automatically generated graphic, and does not have a gendered name that is apparent from the login name. Likewise, we define a gendered profile as one where the gender can be readily inferred from the image or the name. Figure 1 also gives an example of a gendered profile; the profile of "JustinAMiddleton" is gendered because it uses a login name (Justin) commonly associated with men, and because the image depicts a person with masculine features (e.g., pronounced brow ridge (*Brown & Perrett, 1993*)). Clicking on a user's name in pull requests reveals their profile, which may contain more information such as a user-selected display name (like "Justin Middleton").

### Identifiable analysis

To obtain a sample of gendered and gender-neutral profiles, we used a combination of automated and manual techniques. For gendered profiles, we included GitHub users who used a profile image rather than an identicon and that Vasilescu and colleagues' tool could confidently infer a gender from the user's name (*Vasilescu, Capiluppi & Serebrenik, 2014*).

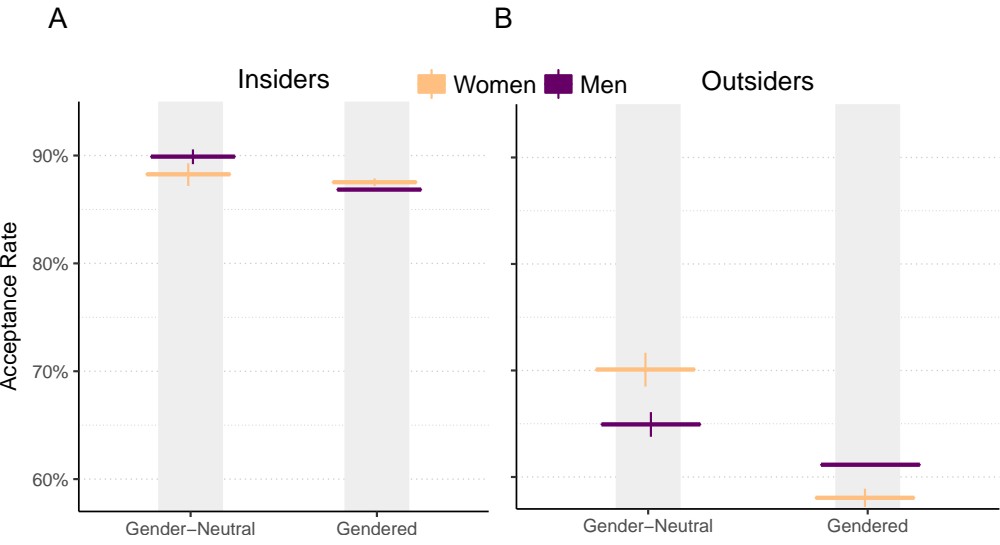

**Figure 6** **Pull request acceptance rate by gender and perceived gender, with 95% Clopper–Pearson confidence intervals, for insiders (A) and outsiders (B).**

For gender-neutral profiles, we included GitHub users that used an identicon, that the tool could not infer a gender for, and that a mixed-culture panel of judges could not guess the gender for.

While acceptance rate results so far have been robust to differences between insiders (people who are owners or collaborators of a project) versus outsiders (everyone else), for this analysis, there is a substantial difference between the two, so we treat each separately. Figure 6 shows the acceptance rates for men and women when their genders are identifiable versus when they are not, with pull requests submitted by insiders on the left and pull requests submitted by outsiders on the right.

### Identifiable results

For insiders, we observe little evidence of bias when we compare women with gender-neutral profiles and women with gendered profiles, since both have similar acceptance rates. This can be explained by the fact that insiders likely know each other to some degree, since they are all authorized to make changes to the project, and thus may be aware of each others' gender.

For outsiders, we see evidence for gender bias: women's acceptance rates drop by 12.0% when their gender is identifiable, compared to when it is not ($\chi^2(df = 1, n = 16,258) = 158, p < .001$). There is a smaller 3.8% drop for men ($\chi^2(df = 1, n = 608,764) = 39, p < .001$). Women have a higher acceptance rate of pull requests overall (as we reported earlier), but when they are outsiders and their gender is identifiable, they have a lower acceptance rate than men.

### Are acceptance rates different if we control for covariates?

In analyses of pull request acceptance rates up until this point, covariates other than the variable of interest (gender) may also contribute to acceptance rates. We have previously

shown an imbalance in covariate distributions for men and women (e.g., number of projects contributed to and number of changes made) and this imbalance may confound the observed gender differences. In this section, we re-analyze acceptance rates while controlling for these potentially confounding covariates using *propensity score matching*, a technique that supports causal inference by transforming a dataset from a non-randomized field study into a dataset that "looks closer to one that would result from a perfectly blocked (and possibly randomized) experiment" (*Ho et al., 2011*). That is, by making gender comparisons between subjects having the same propensity scores, we are able to remove the confounding effects, giving stronger evidence that any observed differences are primarily due to gender bias.

While full details of the matching procedure can be found in the Appendix, in short, propensity score matching works by matching data from one group to similar data in another group (in our case, men's and women's pull requests), then discards the data that do not match. This discarded data represent outliers, and thus the results from analyzing matched data may differ substantially from the results from analyzing the original data. The advantage of propensity score matching is that it controls for any differences we observed earlier that are caused by a measured covariate, rather than gender bias. One negative side effect of matching is that statistical power is reduced because the matched data are smaller than from the original dataset. We may also observe different results than in the larger analysis because we are excluding certain subjects from the population having atypical covariate value combinations that could influence the effects in the previous analyses.

Figure 7 shows acceptance using matched data for all pull requests, for just pull requests from outsiders, and for just pull requests on projects that are open source (OSS) licenses. Asterisks (*) indicate that each difference is statistically significant using a chi-squared test, though the magnitude of the difference between men and women is smaller than for unmatched data.

Figure 8 shows acceptance rates for matched data, analogous to Fig. 5. We calculate statistical significance with a chi-squared test, with a Benjamini–Hochberg correction (*Benjamini & Hochberg, 1995*). For programming languages, acceptance rates for three (Ruby, Python, and C ++) are significantly higher for women, and one (PHP) is significantly higher for men.

Figure 9 shows acceptance rates for matched data by pull request index, that is, for each user's first pull request, second and third pull request, fourth through seventh pull request, and so on. We perform chi-squared tests and Benjamini–Hochberg corrections here as well. Compared to Fig. 3, most differences between genders diminish to the point of non-statistical significance.

From Fig. 9, we might hypothesize that the overall difference in acceptance rates between genders is due to just the first pull request. To examine this, we separate the pull request acceptance rate into:

- **One-Timers**: Pull requests from people who only ever submit one pull request.
- **Regulars' First**: First pull requests from people who go on to submit other pull requests.
- **Regulars' Rest**: All other (second and beyond) pull requests.

 

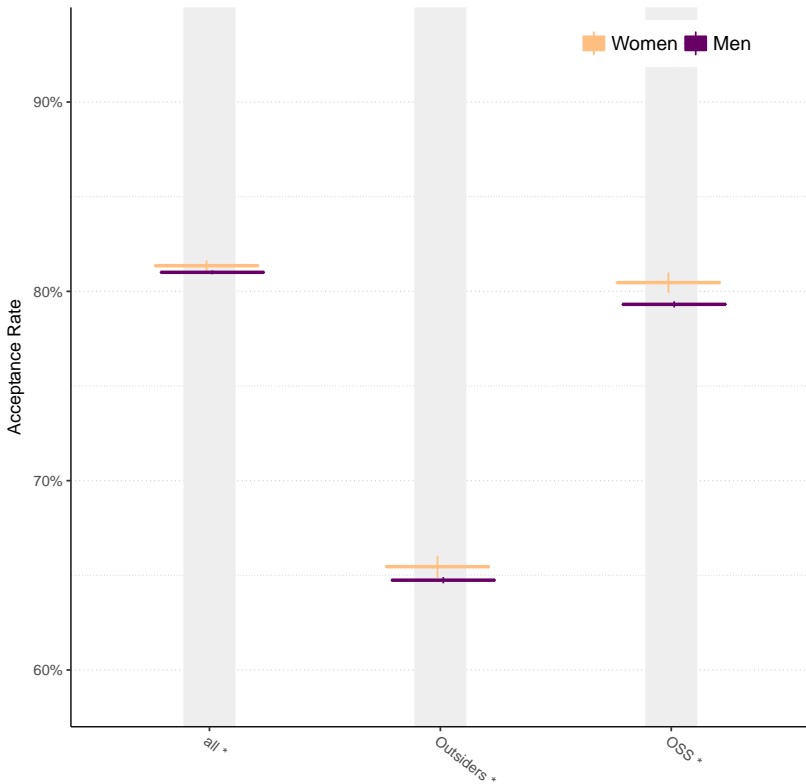

**Figure 7  Acceptance rates for men and women for all data, outsiders, and open source projects using matched data.**

 shows the results. Overall, women maintain a significantly higher acceptance rate beyond the first pull request, disconfirming the hypothesis.

We next investigate acceptance rate by gender and perceived gender using matched data. Here we match slightly differently, matching on identifiability (gendered, unknown, or neutral) rather than use of an identicon. Unfortunately, matching on identifiability (and the same covariates described in this section) reduces the sample size of gender neutral pulls by an order of magnitude, substantially reducing statistical power.[1]

Consequently, here we relax the matching criteria by broadening the equivalence classes for numeric variables.  plots the result.

For outsiders, while men and women perform similarly when their genders are neutral, when their genders are apparent, men's acceptance rate is 1.2% higher than women's ($\chi^2(df = 1, n = 419{,}411) = 7, p < .01$).

How has this matched analysis of the data changed our findings? Our observation about overall acceptance rates being higher for women remains, although the difference is smaller. Our observation about womens' acceptance rates being higher than mens' for all programming languages is now mixed; instead, women's acceptance rate is significantly higher for three languages, but significantly lower for one language. Our observation that womens' acceptance rates continue to outpace mens' becomes less clear. Finally, for outsiders, although gender-neutral women's acceptance rates no longer outpace men's to a

[1] For the sake of completeness, the result of that matching process is included in the Supplemental Files.

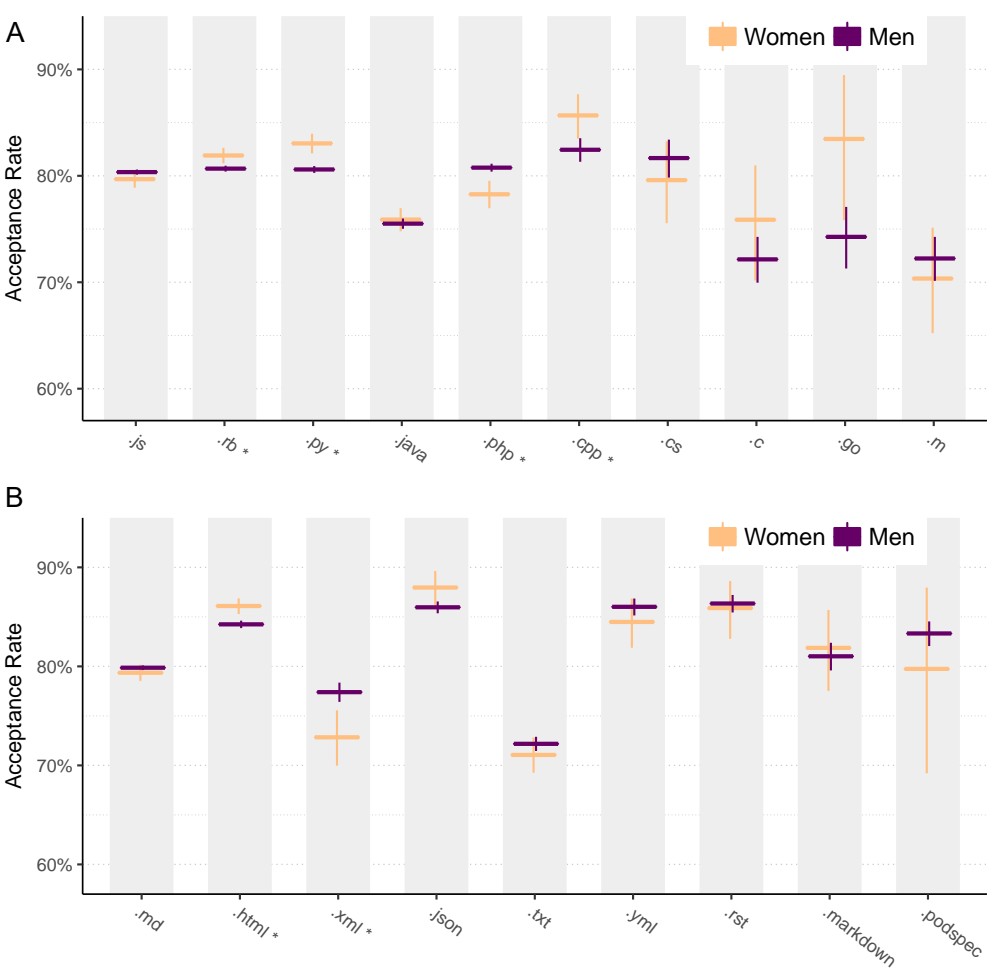

**Figure 8** Acceptance rates for men and women using matched data by file type for programming languages (A) and non-programming languages (B).

statistically significant extent, men's pull requests continue to be accepted more often than women's when the contributor's gender is apparent.

## DISCUSSION

### Why do differences exist in acceptance rates?

To summarize this paper's observations:

1. Women are more likely to have pull requests accepted than men.
2. Women continue to have high acceptance rates as they do pull requests on more projects.
3. Women's pull requests are less likely to serve an documented project need.
4. Women's changes are larger.
5. Women's acceptance rates are higher for some programming languages.
6. Men outsiders' acceptance rates are higher when they are identifiable as men.

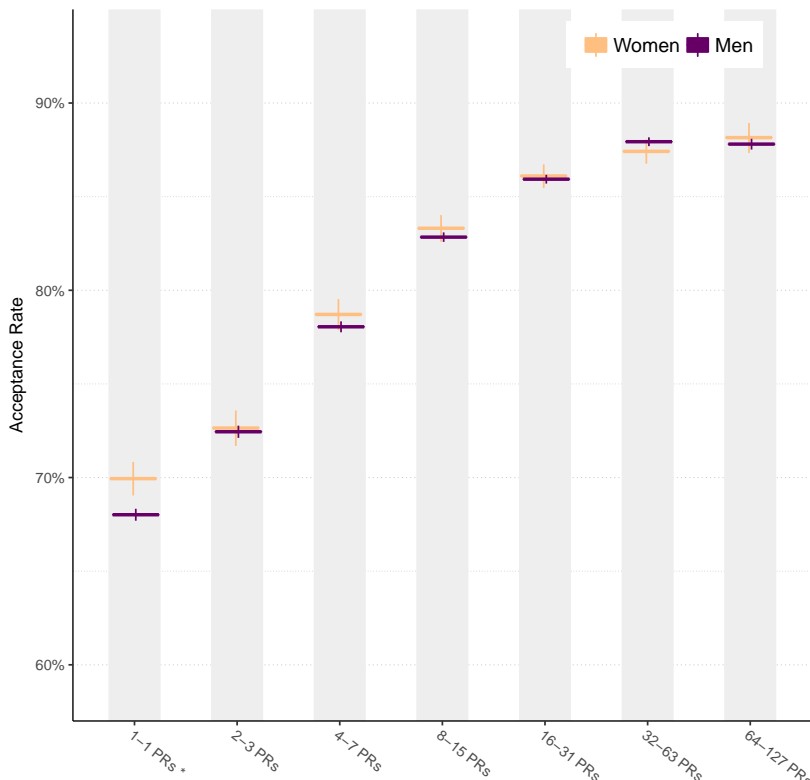

**Figure 9** Pull request acceptance rate over time using matched data.

We next consider several alternative theories that may explain these observations as a whole.

Given observations 1–5, one theory is that a bias against *men* exists, that is, a form of reverse discrimination. However, this theory runs counter to prior work (e.g., *Nafus, 2012*), as well as observations 6.

Another theory is that women are taking fewer risks than men. This theory is consistent with Byrnes' meta-analysis of risk-taking studies, which generally find women are more risk-averse than men (*Byrnes, Miller & Schafer, 1999*). However, this theory is not consistent with observation 4, because women tend to change more lines of code, and changing more lines of code correlates with an increased risk of introducing bugs (*Mockus & Weiss, 2000*).

Another theory is that women in open source are, on average, more competent than men. In Lemkau's review of the psychology and sociology literature, she found that women in male-dominated occupations tend to be highly competent (*Lemkau, 1979*). This theory is consistent with observations 1–5. To be consistent with observations 6, we need to explain why women's pull request acceptance rate drops when their gender is apparent. An addition to this theory that explains observation 6, and the anecdote described in the introduction, is that discrimination against women does exist in open source.

Assuming this final theory is the best one, why might it be that women are more competent, on average? One explanation is survivorship bias: as women continue their formal and informal education in computer science, the less competent ones may change

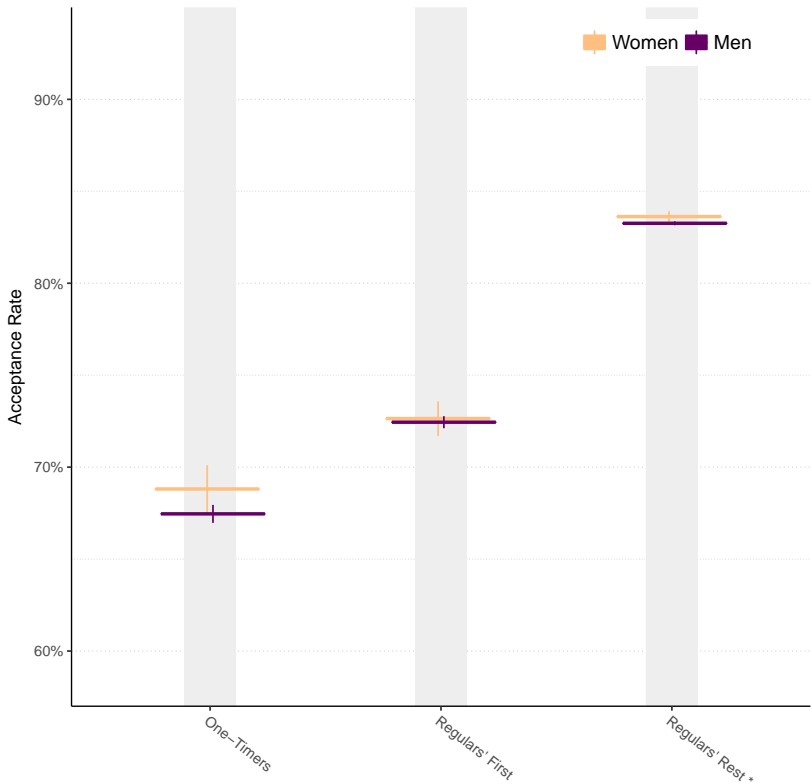

**Figure 10    Acceptance rates for men and women broken down by category.**

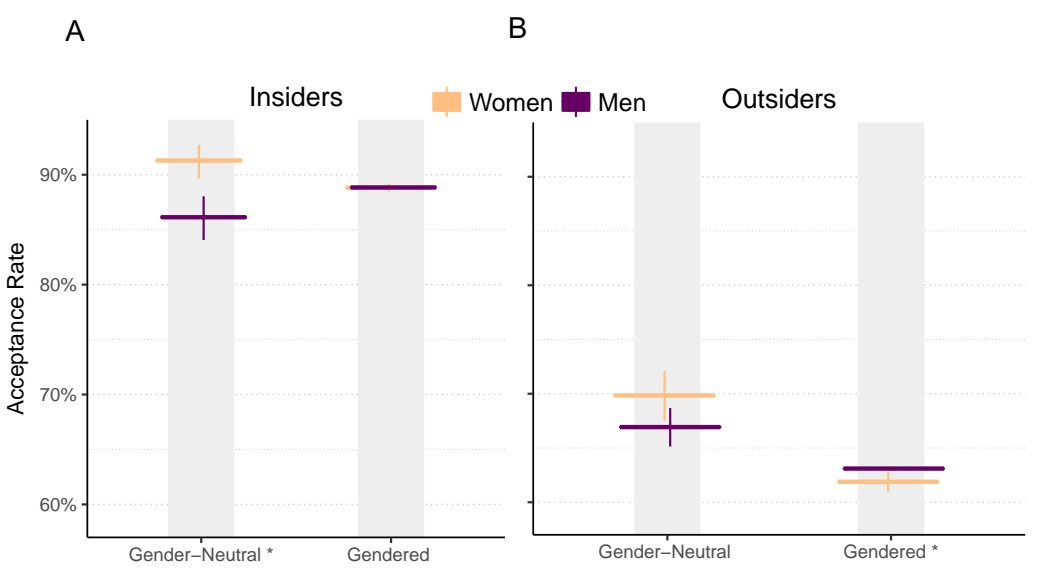

**Figure 11    Pull request acceptance rate by gender and perceived gender, using matched data.**

fields or otherwise drop out. Then, only more competent women remain by the time they begin to contribute to open source. In contrast, less competent men may continue. While women do switch away from STEM majors at a higher rate than men, they also have a lower drop out rate then men (*Chen, 2013*), so the difference between attrition rates of women and men in college appears small. Another explanation is self-selection bias: the average woman in open source may be better prepared than the average man, which is supported by the finding that women in open source are more likely to hold Master's and PhD degrees (*Arjona-Reina, Robles & Dueas, 2014*). Yet another explanation is that women are held to higher performance standards than men, an explanation supported by *Gorman & Kmec (2007)* analysis of the general workforce, as well as Heilman and colleagues' *(2004)* controlled experiments.

### Are the differences meaningful?

We have demonstrated *statistically* significant differences between men's and women's pull request acceptance rates, such as that, overall, women's acceptance rates are 4.1% higher than men's. We caution the reader from interpreting too much from statistical significance; for big data studies such as this one, even small differences can be statistically significant. Instead, we encourage the reader to examine the size of the observed effects. We next examine effect size from two different perspectives.

Using our own data, let us compare acceptance rate to two other factors that correlate with pull request acceptance rates. First, the slope of the lines in Fig. 3, indicate that, generally, as developers become more experienced, their acceptance rates increases fairly steadily. For instance, as experience doubles from 16 to 32 pull requests for men, pull acceptance rate increases by 2.9%. Second, the larger a pull request is, the less likely it is to be accepted (*Gousios, Pinzger & Deursen, 2014*). In our pull request data, for example, increasing the number of files changed from 10 to 20 decreases the acceptance rate by 2.0%.

Using others' data, let us compare our effect size to effect sizes reported in other studies of gender bias. Davison and Burke's meta-analysis of sex discrimination found an average Pearson correlation of $r = .07$, a standardized effect size that represents the linear dependence between gender and job selection (*Davison & Burke, 2000*). In comparison, our 4.1% overall acceptance rate difference is equivalent to $r = .02$.[2] Thus, the effect we have uncovered is only about a quarter of the effect in typical studies of gender bias.

### CONCLUSION

In closing, as anecdotes about gender bias persist, it is imperative that we use big data to better understand the interaction between genders. While our big data study does not prove that differences between gendered interactions are caused by bias among individuals, combined with qualitative data about bias in open source (*Nafus, 2012*), the results are troubling.

Our results show that women's pull requests tend to be accepted more often than men's, yet women's acceptance rates are higher only when they are not identifiable as women. In the context of existing theories of gender in the workplace, plausible explanations include

[2]Calculated using the `chies` function in the `compute.es` R package (https://cran.r-project.org/web/packages/compute.es/compute.es.pdf).

the presence of gender bias in open source, survivorship and self-selection bias, and women being held to higher performance standards.

While bias can be mitigated—such as through "bias busting" workshops (http://www.forbes.com/sites/ellenhuet/2015/11/02/rise-of-the-bias-busters-how-unconscious-bias-became-silicon-valleys-newest-target), open source codes of conduct (http://contributor-covenant.org) and blinded interviewing (https://interviewing.io)— the results of this paper do not suggest which, if any, of these measures should be adopted. More simply, we hope that our results will help the community to acknowledge that biases are widespread, to reevaluate the claim that open source is a pure meritocracy, and to recognize that bias makes a practical impact on the practice of software development.

## ACKNOWLEDGEMENTS

Special thanks to Denae Ford for her help throughout this research project. Thanks to the Developer Liberation Front for their reviews of this paper. For their helpful discussions, thanks to Tiffany Barnes, Margaret Burnett, Tim Chevalier, Aaron Clauset, Julien Couvreur, Prem Devanbu, Ciera Jaspan, Saul Jaspan, David Jones, Jeff Leiter, Ben Livshits, Titus von der Malsburg, Peter Rigby, David Strauss, Bogdan Vasilescu, and Mikael Vejdemo-Johansson. For their helpful critiques during the peer review process, thanks to Lynn Conway, Caroline Simard, and the anonymous reviewers.

## APPENDIX: MATERIALS AND METHODS

### GitHub scraping

An initial analysis of GHTorrent pull requests showed that our pull request merge rate was significantly lower than that presented in prior work on pull requests (*Gousios, Pinzger & Deursen, 2014*). We found a solution to the problem that calculated pull request status using a different technique, which yielded a pull request merge rate comparable to prior work. However, in a manual inspection of pull requests, we noticed that several calculated pull request statuses were different than the statuses indicated on the http://github.com website. As a consequence, we wrote a web scraping tool that automatically downloaded the pull request HTML pages, parsed them, and extracted data on status, pull request message, and comments on the pull request. We performed this process for all pull requests submitted by GitHub users that we had labeled as either a man or woman. In the end, the pull request acceptance rate was 74.8% for all processed pull requests.

We determined whether a pull requestor was an insider or an outsider during our scraping process because the data was not available in the GHTorrent dataset. We classified a user as an insider when the pull request explicitly listed the person as a collaborator or owner (https://help.github.com/articles/what-are-the-different-access-permissions/#user-accounts), and classified them as an outsider otherwise. This analysis has inaccuracies because GitHub users can change roles from outsider to insider and vice-versa. As an example, about 5.9% of merged pull requests from both outsider female and male users were merged by the outsider pull-requestor themselves, which is not possible, since outsiders

by definition do not have the authority to self-merge. We emailed such an outsider, who indicated that, indeed, she was an insider when she made that pull request. We attempted to mitigate this problem by using a technique similar to that used in prior work (*Gousios, Pinzger & Deursen, 2014*; *Yu et al., 2015*). From contributors that we initially marked as outsiders, for a given pull request on a project, we instead classified them as insiders when they met any of three conditions. The first condition was that they had closed an issue on the project within 90 days prior to opening the given pull request. The second condition was that they had merged the given pull request or any other pull request on the project in the prior 90 days. The third condition was that they had closed any pull request that someone else had opened in the prior 90 days. Meeting any of these conditions implies that, even if the contributor was an outsider at the time of our scraping, they were probably an insider at the time of the pull request.

## Gender linking

To evaluate gender bias on GitHub, we first needed to determine the genders of GitHub users.

Our technique uses several steps to determine the genders of GitHub users. First, from the GHTorrent data set, we extract the email addresses of GitHub users. Second, for each email address, we use the search engine in the Google+ social network to search for users with that email address. The search works for both Google users' email addresses (@gmail.com), as well as other email addresses (such as @ncsu.edu). Third, we parse the returned users' 'About' page to scrape their gender. Finally, we include only the genders 'Male' and 'Female' (334,578 users who make pull requests) because there were relatively few other options chosen (159 users). We also automated and parallelized this process. This technique capitalizes on several properties of the Google+ social network. First, if a Google+ user signed up for the social network using an email address, the search results for that email address will return just that user, regardless of whether that email address is publicly listed or not. Second, signing up for a Google account currently *requires* you to specify a gender (though 'Other' is an option) (https://accounts.google.com/SignUp), and, in our discussion, we interpret their use of 'Male' and 'Female' in gender identification (rather than sex) as corresponding to our use of the terms 'man' and 'woman'. Third, when Google+ was originally launched, gender was publicly visible by default (http://latimesblogs.latimes.com/technology/2011/07/google-plus-users-will-soon-be-able-to-opt-out-of-sharing-gender.html).

## Merged pull requests

Throughout this study, we measure pull requests that are accepted by calculating developers' merge rates, that is, the number of pull requests merged divided by the sum of the number of pull requests merged, closed, and still open. We include pull requests still open in the denominator in this calculation because pull requests that are still open could be indicative of a pull requestor being ignored, which has the same practical impact as rejection.

## Project licensing

To determine whether a project uses an open source license, we used an experimental GitHub API that uses heuristics to determine a project's license (https://developer.github.com/v3/licenses/). We classified a project (and thus the pull request on that project) as open source if the API reported a license that the Open Source Initiative considers in compliance with the Open Source Definition (https://opensource.org/licenses), which were afl-3.0, agpl-3.0, apache-2.0, artistic-2.0, bsd-2-clause, bsd-3-clause, epl-1.0, eupl-1.1, gpl-2.0, gpl-3.0, isc, lgpl-2.1, lgpl-3.0, mit, mpl-2.0, ms-pl, ms-rl, ofl-1.1, and osl-3.0. Projects were not considered open source if the API did not return a license for a project, or the license was bsd-3-clause-clear, cc-by-4.0, cc-by-sa-4.0, cc0-1.0, other, unlicense, or wtfpl.

## Determining gender neutral and gendered profiles

To determine gendered profiles, we first parsed GitHub profile pages to determine whether each user was using a profile image or an identicon. Of the users who performed at least one pull request, 213,882 used a profile image and 104,648 used an identicon. We then ran display names and login names through a gender inference program, which maps a name to a gender.[3] We classified a GitHub profile as gendered if each of the following were true:

- a profile image (rather than an identicon) was used, and
- the gender inference tool output a gender at the highest level of confidence (that is, 'male' or 'female,' rather than 'mostly male,' 'mostly female,' or 'unknown').

We classified profile images as identicons using ImageMagick (http://www.graphicsmagick.org/GraphicsMagick.html), looking for an identicon-specific file size, image dimension, image class, and color depth. In an informal inspection into profile images, we found examples of non-photographic images that conveyed gender cues, so we did not attempt to distinguish between photographic and non-photographic images when classifying profiles as gendered.

To classify profiles as gender neutral, we added a manual step. Given a GitHub profile that used an identicon (thus, a gender could not be inferred from a profile image) and a name that the gender inference tool classified as 'unknown', we manually verified that the profile could not be easily identified as belonging to a specific gender. We did this in two phases. In the first phase, we assembled a panel of 3 people to evaluate profiles for 10 s each. The panelists were a convenience sample of graduate and undergraduate students from North Carolina State University. Panelists were of American (man), Chinese (man), and Indian (woman) origin, representative of the three most common nationalities on GitHub. We used different nationalities because we wanted the panel to be able to identify, if possible, the genders of GitHub usernames with different cultural origins. In the second phase, we eliminated two inefficiencies from the first phase: (a) because the first panel estimated that for 99% of profiles, they only looked at login names and display names, we only showed this information to the second panel, and (b) because the first panel found 10 s was usually more time than was necessary to assess gender, we allowed panelists at the second phase to assess names at their own pace. Across both phases, panelists were instructed to signal if they could identify the gender of the GitHub profile. To estimate

[3]This tool was builds on Vasilescu and colleagues' tool (*Vasilescu, Capiluppi & Serebrenik, 2014*), but we removed some of Vasilescu and colleagues' heuristics to be more conservative. Our version of the tool can be found here: https://github.com/DeveloperLiberationFront/genderComputer.

panelists' confidence, we considered using a threshold like "90% confident of the gender," but found that this was too ambiguous in pilot panels. Instead, we instructed panelists to signal if they would be comfortable addressing the GitHub user as 'Mister' or 'Miss' in an email, given the only thing they knew about the user was their profile. We considered a GitHub profile as gender neutral if all of the following conditions were met:

- an identicon (rather than a profile image) was used,
- the gender inference tool output a 'unknown' for the user's login name and display name, and
- none of the panelists indicated that they could identify the user's gender.

Rather than asking a panel to laboriously evaluate every profile for which the first two criteria applied, we instead asked panelists to inspect a random subset. Across both panels, panelists inspected 3,000 profiles of roughly equal numbers of women and men. We chose the number 3,000 by doing a rough statistical power analysis using the results of the first panel to determine how many profiles panelists should inspect during the second panel to obtain statistically significant results. Of the 3,000, panelists eliminated 409 profiles for which at least one panelist could infer a gender.

## Matching procedure

To enable more confident causal inferences about the effect of gender, we used propensity score matching to remove the effect of confounding factors from our acceptance rate analyses. In our analyses, we used men as the control group and women as the treatment group. We treated each pull request as a data point. The covariates we matched were number of lines added, number of lines removed, number of commits, number of files changed, pull index (the creator's $n$th pull request), number of references to issues, license (open source or not), creator type (owner, collaborator, or outsider), file extension, and whether the pull requestor used an identicon. We excluded pull requests for which we were missing data for any covariate.

We used the R library MatchIt (*Ho et al., 2011*). Although MatchIt offers a variety of matching techniques, such as full matching and nearest neighbor, we found that only the exact matching technique completed the matching process, due to our large number of covariates and data points. With exact matching, each data point in the treatment group must match exactly with one or more data points in the control group. This presents a problem for covariates with wide distributions (such as lines of code) because it severely restricts the technique's ability to find matches. For instance, if a woman made a pull request with 700 lines added and a man made a pull request with 701 lines added that was otherwise identical (same number of lines removed, same file extension, and so on), these two data points would not be matched and excluded from further analysis. Consequently, we pre-processed each numerical variable into the floor of the log2 of it. Thus, for example, both 700 and 701 are transformed into 5, and thus can be exactly matched.

After exact matching, the means of all covariates are balanced, that is, their weighted means are equal across genders. Raw numerical data, since we transformed it, is not

perfectly balanced, but is substantially more balanced than the original data; each covariate showed a 96% or better balance improvement.

Finally, as we noted in the matching procedure for gendered and gender-neutral contributors, to retain reasonable sample sizes, we relaxed the matching criteria by broadening the equivalence classes for numeric variables. Specifically, for lines added, lines removed, commits, files changed, pull index, and references, we transformed the data using log10 rather than log2.

## Missing data

In some cases, data were missing when we scraped the web to obtain data to supplement the GHTorrent data. We describe how we dealt with these data here.

First, information on file types was missing for pull requests that added or deleted more than 1,000 lines. The problem was that GitHub does not include file type data on initial page response payloads for large changes, presumably for efficiency reasons. This missing data affects the results of the file type analysis and the propensity score matching analysis; in both cases, pull requests of over 1,000 lines added or deleted are excluded.

Second, when retrieving GitHub user images, we occasionally received abnormal server response errors, typically in the form of HTTP 404 errors. Thus, we were unable to determine if the user used a profile image or identicon in 10,458 (3.2% of users and 2.0% of pull requests). We excluded these users and pull requests when analyzing data on gendered users.

Third, when retrieving GitHub pull request web pages, we occasionally received abnormal server responses as well. In these cases, we were unable to obtain data on the size of the change (lines added, files changed, etc.), the state (closed, merged, or open), the file type, or the user who merged or closed it, if any. This data comprises 5.15% of pull requests for which we had genders of the pull request creator. These pull requests are excluded from all analyses.

## Threats

One threat to this analysis is that additional covariates, including ones that we could not collect, may influence acceptance rate. One example is that we did not account for the GitHub user judging pull requests, even though such users are central to the pull request process. Another example is pull requestors' programming experience outside of GitHub. Two covariates we collected, but did not control for, is the project the pull request is made to and the developer deciding on the pull request. We did not control for these covariates because we reasoned that it would discard too many data points during matching.

Another threat to this analysis is the existence of robots that interact with pull requests. For example, "Snoopy Crime Cop" (https://github.com/snoopycrimecop) appears to be a robot that has made thousands of pull requests. If such robots used an email address that linked to a Google profile that listed a gender, our merge rate calculations might be skewed unduly. To check for this possibility, we examined profiles of GitHub users that we have genders for and who have made more than 1,000 pull requests. The result was tens of GitHub users, none of whom appeared to be a robot. So in terms of our merge calculation, we are somewhat confident that robots are not substantially influencing the results.

Another threat is if men and women misrepresent their genders. If so, we inaccurately label some men on GitHub as women, and vice-versa. While emailing the thousands of pull requestors described in this study to confirm their gender is feasible, doing so is ethically questionable; GHTorrent no longer includes personal email addresses, after GitHub users complained of receiving too many emails from researchers (https://github.com/ghtorrent/ghtorrent.org/issues/32).

Another threat is GitHub developers' use of aliases (*Vasilescu et al., 2015*); the same person may appear as multiple GitHub users. Each alias artificially inflates the number of developers shown in the histograms in Fig. 2. Most pull request-level analysis, which represents most of the analyses performed in this paper, are unaffected by aliases that use the same email address.

Another threat is inaccuracies in our assessment of whether a GitHub member's gender is identifiable. First, the threat in part arises from our use the genderComputer program. When genderComputer labels a GitHub profile as belonging to a man, but a human would perceive the user's profile as belonging to a woman (or vice-versa), then our classification of gendered profiles is inaccurate in such cases. We attempted to mitigate this risk by discarding any profiles in the gendered profile analysis that genderComputer classified with low-confidence. Second, the threat in part arises from our panel. For profiles we labeled as gender-neutral, our panel may not have picked out subtle gender features in GitHub users' profiles. Moreover, project owners may have used gender signals that we did not; for example, if a pull requestor sends an email to a project owner, the owner may be able to identify the requestor's gender even though our technique could not.

A similar threat is that users who judge pull requests encounter gender cues by searching more deeply than we assume. We assume that the majority of users judging pull requests will look only at the pull request itself (containing the requestor's username and small profile image) and perhaps the requestor's GitHub profile (containing username, display name, larger profile image, and GitHub contribution history). Likewise, we assume that very few users judging pull requests will look into the requestor further, such as into their social media profiles. Although judges could have theoretically found requestors' Google+ profiles using their email addresses (as we did), this seems unlikely for two reasons. First, while pull requests have explicit links to a requestor's GitHub profile, they do not have explicit links to a requestor's social media profile; the judge would have to instead seek them out, possibly using a difficult-to-access email address. Second, we claim that our GitHub-to-Google+ linking technique is a novel research technique; assuming that it is also novel in practice, users who judge pull requests would not know about it and therefore would be unable to look up a user's gender on their Google+ profile.

Another threat is that of construct validity, whether we are measuring what we aim to measure. One example is our inclusion of "open" pull requests as a sign of rejection, in addition to the "closed" status. Rather than a sign of rejection, open pull requests may simply have not yet been decided upon. However, these pull requests had been open for at least 126 days, the time between when the last pull request was included in GHTorrent and when we did our web scrape. Given Gousios and colleagues' *(2014)* finding that 95% of pull requests are closed within 26 days, insiders likely had ample time to decide on open

**Table 1** **Acceptance rates for GitHub users not linked to Google+ (top row) versus those who are linked (bottom rows), by stated gender.** Right three columns indicate the percentiles of the number of projects contributed to.

| Gender category | Users | Pull requests | Acceptance rate | 95% Confidence interval |
|---|---|---|---|---|
| User not on Google+ | 325,100 | 3,047,071 | 71.5% | [71.44%,71.54%] |
| User identifies as 'Male' on Google+ | 312,909 | 3,168,365 | 74.2% | [74.17%,74.27%] |
| User identifies as 'Female' on Google+ | 21,510 | 156,589 | 79.9% | [79.69%,80.09%] |
| User has no gender listed on Google+ | 20,024 | 194,837 | 74.3% | [74.09%,74.48%] |
| User lists 'Declined to State' for gender on Google+ | 7,484 | 81,632 | 73.1% | [72.80%,73.41%] |
| User lists other gender on Google+ | 159 | 1,339 | 73.9% | [71.50%,76.27%] |

**Table 2** **Percentiles of the number of projects contributed to for GitHub users not linked to Google+ (top row) versus those who are linked (bottom rows), by stated gender.**

| Gender category | Users | Pull requests | 50% | 75% | 90% |
|---|---|---|---|---|---|
| User not on Google+ | 325,100 | 3,047,071 | 1.00 | 3.00 | 6.00 |
| User identifies as 'Male' on Google+ | 312,909 | 3,168,365 | 1.00 | 3.00 | 7.00 |
| User identifies as 'Female' on Google+ | 21,510 | 156,589 | 1.00 | 2.00 | 4.00 |
| User has no gender listed on Google+ | 20,024 | 194,837 | 1.00 | 3.00 | 7.00 |
| User lists 'Declined to State' for gender on Google+ | 7,484 | 81,632 | 1.00 | 3.00 | 7.00 |
| User lists other gender on Google+ | 159 | 1,339 | 2.00 | 4.00 | 7.20 |

pull requests. Another example is whether pull requests that do not link to issues signals that the pull request does not fulfill a documented need. A final example is that a GitHub user might be an insider without being an explicit owner or collaborator; for instance, a user may be well-known and trusted socially, yet not be granted collaborator or owner status, in an effort to maximize security by minimizing a project's attack surface (*Howard, Pincus & Wing, 2005*).

Another threat is that of external validity; do the results generalize beyond the population studied? While we chose GitHub because it is the largest open source community, other communities such as SourceForge and BitBucket exist, along with other ways to make pull requests, such at through the git version control system directly. Thus, our study provides limited generalizability to other open source ecosystems. Moreover, while we studied a large population of contributors, they represent only part of the total population of developers on GitHub, because not every developer makes their email address public, because not every email address corresponds to a Google+ profile, and because not every Google+ profile lists gender.

To understand this threat, Tables 1 and 2 compare GitHub users who we could link to Google+ accounts (the data we used in this paper) against those who do not have Google+ accounts. The top 3 rows are the main ones of interest. In Table 1, we use an exclusively GHTorrent-based calculation of acceptance rate where a pull request is considered accepted if its commit appears in the commit history of the project; we use a different measure of acceptance rate here because we did not parse pull requests made by people not on Google+.

In terms of acceptance rate, users *not* on Google+ have a lower acceptance rate than both males and females on Google+. In terms of number of unique projects contributed to, users not on Google+ contribute to about the same number as men on Google+.

A final threat to this research is our own biases as researchers, which may have influenced the results. While it is difficult to control for implicit bias, we can explicitly state what our biases are, and the reader can interpret the findings in that context. First, prior to conducting this research, all researchers on the team did believe that gender bias exists in open source communities, based on personal experience, news articles, and published research. However, none knew how widespread it was, or whether that bias could be detected in pull requests. Second, all researchers took Nosek and colleagues' *(2002)* online test for implicit bias that evaluates a person's implicit associations between males and females, and work and family. As is typical with most test takers, most authors tended to associate males with work and females with family (Kofink: strong; Murphy-Hill, Parnin, and Stallings: moderate; Terrell and Rainear: slight). The exception was Middleton, who exhibits a moderate association of female with career and male with family.

### Funding
This material is based in part upon work supported by the National Science Foundation under grant number 1252995. There was no additional external funding received for this study. The funders had no role in study design, data collection and analysis, decision to publish, or preparation of the manuscript.

### Grant Disclosures
The following grant information was disclosed by the authors:
National Science Foundation: 1252995.

### Competing Interests
The authors declare there are no competing interests.

### Author Contributions
- Josh Terrell, Andrew Kofink and Emerson Murphy-Hill conceived and designed the experiments, analyzed the data, contributed reagents/materials/analysis tools, wrote the paper, prepared figures and/or tables, performed the computation work, reviewed drafts of the paper.
- Justin Middleton conceived and designed the experiments, analyzed the data, contributed reagents/materials/analysis tools, wrote the paper, reviewed drafts of the paper.
- Clarissa Rainear analyzed the data, wrote the paper, reviewed drafts of the paper.
- Chris Parnin conceived and designed the experiments, contributed reagents/materials/analysis tools, wrote the paper, prepared figures and/or tables, reviewed drafts of the paper.
- Jon Stallings conceived and designed the experiments, analyzed the data, wrote the paper, reviewed drafts of the paper.
## Ethics

The following information was supplied relating to ethical approvals (i.e., approving body and any reference numbers):

NCSU IRB approved under #6708.

## Data Availability

Data sets from GHTorrent and Google+ are publicly available. Raw data from figures has been supplied as a Supplementary File.

## Supplemental Information

Supplemental information for this article can be found online at http://dx.doi.org/10.7717/peerj-cs.111#supplemental-information.

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
