# Peer review of "Gender differences and bias in open source: pull request acceptance of women versus men"

_PeerJ Computer Science, doi:10.7717/peerj-cs.111_

## Round 0.1 · original submission · Major Revisions

Dear authors,

Thank you again for submitting your manuscript to PeerJ. Our apologies for taking more time than normal for the reviewing process. We hope that the quality of the resulting reviews makes up for this delay.

All reviewers agree that the authors present an impressive and important statistical analysis relating gender identity and pull request acceptance.

Reviewer 1 expresses concerns that most of the write up of the findings does not address the core research question about bias. Reviewer 1 suggests a reorganization of the material that could strengthen the connection with the primary research question, and which would also help to articulate the points of the insider and competence effects.

Reviewer 2 suggests a number of small changes that help to avoid mis-interpretation of the paper's key results.

Reviewer 3 indicates that certain key elements of the approach require clarification, in particular the identification of non-identifiable women, the identification of insiders, and the use of propensity score matching.

Reviewer 4 raises concerns that seem related to those of Reviewer 1, namely the connection of the findings with bias. The comments by Reviewer 4 call for a critical explanation to what extent the results can be truly attributed to bias.

Reviewer 4 also missed an outlook of what can be done with the results: Could the paper discuss to what actions these results might lead?

All in all the reviews call for the following:

1. Consider whether a re-ordering of the material as suggested by reviewer 1 helps to get the message better across
2. Explicitly discuss, maybe in a separate discussion subsection, how the presented empirical evidence leads to the conclusion of bias (or how such a conclusion could be drawn)
3. Consider discussing potential implications / actionability of the results (e.g., what can projects do? what can open source contributors do?)

I also was surprised that an otherwise strong and compelling paper ends with a somewhat weak five line conclusion section. This seems like the ideal place to clearly articulate the key findings of the paper.

The appendices contain important material. I am not sure why they are put as an appendix, but I do consider them as an integral part of the paper that needs to be included in the final version of the paper as well.

The reviews don't call for additional research or experiments. I hope and trust that the authors will be able to address the various concerns expressed, which mostly amount to strengthening the analysis and presentation.

Note that Reviewer 1 has identified herself, and has offered the possibility to talk about the paper. This may also be helpful for addressing the concerns expressed by reviewer 4.

I look forward to receiving the revised manuscript and author response to these suggestions!

Arie van Deursen

·

Basic reporting

The paper is clearly written. The literature pertaining to the main finding of the paper, that is the existence of gender bias, and the larger context of women in computing, is coherent and relevant. Figures are relevant and easy to interpret.

Experimental design

This study is indeed one of the largest field data source on gender bias. Few studies exist that pertain to “real world” contexts and go beyond testing the existence of bias in an experimental settings. As such, this paper provides a significant contribution to the literature, and is beneficial to the literature.

Validity of the findings

The data are unique, and at an unprecedented scale, offering significant validity. The research question is relevant, meaningful, and well designed. The investigation is rigorous and the methods well described. One issue to correct is to more clearly link the main research question to the analysis. Most of the write up of the findings do not address the main research question – that is, the question stated on page 5 “to what extent does gender bias exist among people who judge GH requests” is not answered until page 23. Most of the analysis is spent answering the question “why are women’s contributions accepted at higher rates than men”. I would recommend re-organizing the write up of the findings – start with the evidence on bias, then ask the question about women’s higher acceptance rate when bias is removed (e.g. when the gender is unknown to the reviewer), controlling for the appropriate factors and exploring why that might be. We also do not know much in the paper about the people who judge GH requests, although they are central to the research question. This limitation should be noted.

Additional comments

Note: my comments come from the perspective of social science – therefore, my recommendations on methods and specific literature to look to may or may not perfectly match the computing literature and approaches.

This paper makes a significant contribution to the literature by providing large-scale real world evidence of gender bias in Open Source. These findings have significant implications for the design of open source communities and technology workplaces and should be published.

There is a flaw in the way the paper is constructed that warrants a rewrite/re-organization but I think is absolutely achievable:

- The paper starts with a research question on the prevalence of bias in OS, and then spends almost all of its analysis on trying to explain a “surprising” finding that women’s contributions are rated higher when bias is removed. This leaves the reader wondering why they are spending so much time answering what seems to be a different question. The finding that clearly answers the research question, is found in Figure 11: that is, the paper demonstrates that women's contributions are significantly more likely to be accepted when their gender is masked (consistent with the literature), and that the effect is worst for outsiders (which is an important contribution to the literature on how "insider status" may mitigate bias).
- I would recommend starting the analysis with answering the main research question - to what extent is bias present in women's OS contributions? Show the findings from Figure 11th. Then, pose the follow up question "Why are women's acceptance rates so much higher than men's in the gender neutral condition – what could explain this?" - Then, posit one series of hypotheses that could explain the difference and present a single statistical model with all the control variables as opposed to a series of single variable cotrols (if possible). The one by one analysis does not offer one model to control for all variables and understanding the relative impact of one variable compared to another. Then, go into the further delving of the differences by over time data and language.
- The discussion can also strengthen articulating the theoretical contribution of the paper by elaborating on 1) the mitigating effect of being insiders on bias, and 2) the finding that women may be more competent in this context and link back to the social science literature that demonstrates that women have to work harder to be perceived as competent (see the work of Elizabeth Gorman and Julie Kmec for example), especially in masculine domains, and therefore explaining those women who participate in Open Source are indeed likely to be more prepared.

A final comment for the editors and authors: it is commendable and critically important to have multidisciplinary perspectives and approaches to address critical social issues. I feel like it is absolutely within reach for the paper to solidify its ties and contributions to the social science literature while preserving and highlighting the power of computational approaches to analysis.

Happy to speak live if authors have questions or want to discuss ideas more at length (csimard@stanford.edu)

·

Basic reporting

I have found three minor wording problems that could lead to misunderstandings or misinterpretations, and have suggested minor revisions to avoid such difficulties.

1. Regarding this sentence in the abstract (p.1):
“This study presents the largest study to date on gender bias, where we compare acceptance rates of contributions from men versus women in an open source software community”

Taken as a whole, the overall sentence can be interpreted as being limited to a study of bias in an open-source software community. However, the initial phrase “the largest study to date” undermines such interpretation, and is also refuted by prior large scale demographic studies of gender bias in women’s vs men’s earnings, etc. These difficulties/ambiguities can be avoided (see related issue in 2.) by the following minor revision:

Change: “This study presents the largest study to date on gender bias, . . . ”
To: “This study presents a large scale study on gender bias, . . . ”

2. Regarding this sentence at the bottom of p2:
“To our knowledge, this is the largest scale study of gender bias to date.”

This sentence, if/when taken out of context (by media, for example), appears to make an unsupportable claim (due to prior large-scale studies of men’s vs women’s earnings, etc). However, within the context of open-source studies the statement is apparently valid. The difficulty here can be avoided via the following minor revision:

Change: “. . . largest scale study of gender bias to date.”
To: “. . . largest scale study to date of gender bias in open source communities.”

3. Regarding the lead sentence in the concluding paragraph on p6:
“As an aside, we believe that our gender linking approach raises privacy concerns, which we have taken several steps to address.”

This wording weakens an important awareness and response, which can be avoided via the following minor revision:

Change: “As an aside, we believe that our gender linking approach . . . ”
To: “We recognize that our gender linking approach . . . ”

Experimental design

No Comments

Validity of the findings

No Comments

Reviewer 3 ·

Basic reporting

Summary: This article is absolutely interesting! The study investigates the
rate of pull request acceptance for women vs. men. This is the largest study
to date on gender bias and it also has much significance on the impact of
gender bias in the computer science workforce. One of the key results is that
women's pull requests are accepted at a higher rate, which the authors explain
based on several theories on gender bias.

Experimental design

+ The study uses almost 5 years worth of pull requests from GHTorrent data set. This is a very large set of data to draw insights from. Therefore the findings from this study carry much weight and make a strong impact in the area of gender bias studies.

- One of the significant result is that, "Women outsiders' acceptance rates are higher but, only when they are not identifiable as women." This part of the methodology is very confusing, which can impact the validity of this finding. In particular, the reviewer was not able to understand how the authors identified a group of female github users who are not apparently identifiable as women. I understand that the researchers used the method of linking a user's email address to the Google+ social network and extracting the gender information from the Google+ site. However, isn't this something that other users of github could have done for users, where they only see identicons and they need to handle their pull requests? Therefore, I am confused how the study was able to identify a group of "true female contributors" who are not easily identifiable as women.

Along this line, page 17 mentions that "a mixed culture panel of judges could not guess the gender for."-- please explain who are "mixed culture panel of judges" and how they came to consensus. I think it would have been very useful to know whether there was a manual investigation of "gender" by contacting the github users directly through emails to confirm their gender.

- On page 18, there is an analysis of "insiders" vs. "outsiders" -- this part is confusing. How do the authors classify "insiders" vs. "outsiders"? The text mentions "who are owners or contributors" vs. "everyone else" but is this distinction clear cut for every open source project out there on the github? Some company projects hosted on the github have many "insiders" a.k.a contributors in the same company who may not be given the role/ permission of "contributors." Please clarify the classification process in detail.

- I hope you can add some more detailed process of "controlling co-variates" with a concrete example. There's a citation to "propensity score matching" but the reader is not able to understand how the authors identify "similar" data and what is the notion of "similarity" in the subsequent analysis for each dimension? I also have a related question. Does this analysis require the size of data from one group to be the same as the size of another similar group? What if the size of similar groups are very different? The section on page 19 talks about "matched data" but it is not clear how "matched data" are identified.

Validity of the findings

+ The results are definitely interesting and the authors do very good job leading the audience from one question to the next investigation question and interpreting the results in the context of rich literature on gender bias. For example, it starts with the high level question of "are women's pull requests less likely to be accepted" to the question of "do women's pull request acceptance rates start low and increase over time?" I also appreciate the author's effort of mitigating the internal validity of the study by contrasting group of users with similar characteristics.

+ The related work was an excellent introduction to those who are not familiar gender bias studies.

+ The reader very much appreciated the authors' effort of interpreting the results in the context of many theories on gender bias. For example, the theory of the theory of "survivorship bias" and "self selection bias" could explain the observation very well. I also agree that "women are often held to higher performance standards than men" and therefore expected to submit higher quality of contributions.

- The section on "are the differences meaningful?" seems very important as any study with large participants can easily find a statistically significant difference. The part that is very confusing about this section is that the methodology is written at a very high level. I suggest the authors to add some more details on the "effect size" analysis method. For example, "Davison and Burke's meta-analysis of sex discrimination found an average Pearson correlation of r = .07, and the study's overall 4.1% acceptance rate difference is equivalent to r = .02. How does "4.1%" map to r=.02. How should the readers interpret r value? What is the overall conclusion of this number
on the significance of the entire study?

Additional comments

See above. Very interesting work!

Reviewer 4 ·

Basic reporting

The paper presents a study of pull request acceptance differences between genders. The authors ask the question of whether differences exist between (self proclaimed) male and female among GitHub users and find a statistically significant difference of around 4% between the acceptance rates. They then identify several confounding factors and attempt to explain this difference; they find that women's PRs are larger and are less likely to serve a project need and that women continue to have larger acceptance even as their tenure in the project increases. However, when they are outsiders in a project (typical case in GitHub OSS), and their gender is identifiable through their profile pics or name, they tend to be less successful than men. The authors attribute this last difference to the existence of bias against women in OSS.

The paper is written in a straightforward manner that is very refreshing; without too much fanfare, the authors convey large amounts of information. I really enjoyed reading this paper. I also think the plots and the tables in the paper convey all information necessary. To make it short, I believe that in terms of presentation this paper is very good.

The authors chose not to redistribute the data due to privacy concerns. I understand this decision. However, this limits my ability as a reviewer to check the code that generated the data and analysis scripts. In this case, I think it is very important to ensure data quality through transparency. I would strongly recommend to the authors to release the tools and analysis scripts they used.

Experimental design

I have no general comments about the experimental design; it looks like solid science done on an almost full population study. The comments below reflect specific issues I had with parts of the analysis.

- l17: Why is under-representation an illustration of bias? If there are population differences on sites with gender-oriented content would that constitute bias (active discrimination) or different content choices between genders? Please find a reference to support this claim.

- l115: I do not see the connection between interviewee evaluation and PR evaluation. At the very least, the interviewee gender is immediately apparent, which is not true for PRs (as you also show).

- l120: The Clopper-Pearson method assumes a Bernoulli process where the chance of success is constant. I am not sure this is the case here, as the differentiated variable (PR acceptance) varies significantly across projects. Can you justify your choice of test here, or select a non-parametric test such as the Matt-Witney-Wilcoxon rank sum (also to test for significance as in l123)?

- l123 and elsewhere: You test for significance, but you do not report effect sizes. Can you use an appropriate test (such as Cliffs delta) and report its results?

- l294-onwards: I appreciate the fact that you used propensity analysis here; this is novel in software engineering. However, when you use matched data, the differences the main effect you are observing (e.g. differences in the acceptance rate for women when the gender is obvious) vanishes (comparing Fig 7 and 11). Moreover, it looks like (Fig 11) that the drop in acceptance rate for women and increase for men is within the confidence interval of the gendered-neutral case for both genders.

What does this indicate? In my view, that controlling for covariates, there is a trivial difference (if any) in the acceptance rate of PRs between genders.

Materials and methods section: Given that the differences between acceptance rates are very small, it is important to report ALL statistics when describing data analysis methods.

- l493-500: What is the PR acceptance rate after applying all treatments? -

- l501-510: Gousios et al. in ref [13] and Yu et al. in their MSR 2015 paper calculate the core team as the number of people that have either committed directly or merged a PR the last 3 months before the specific processed PR. If you keep track of project core team members for each PR, you can detect members that were added or removed to the core team by looking ahead in time. So if user A was in the core team for PR 12 but not for PRs 13-last PR, you can definitely say that PR is an outsider for PRs 13 onwards.

- l511: What is the technique? This is crucial to report as gives an indication of the coverage of your study over the general population.

- l518: quantify "relatively few"

- l527: Gousios et al. found that 95% of PRs are closed in 25 days; you could have removed open pull requests that are older than 25 days from your denominator.

- l543: How did you determine whether somebody has a profile image rather than an identicon by parsing the source page? How about profile images that use other types of computer graphics (e.g. cartoons, logos etc)?

- l546: Did you check the accuracy of genderComputer? If yes, what is the recall and precision? How did your changes improve those? In general, how can you be sure that what genderComputer calculates is valid?

- l547-578: This is an interesting approach. How did you scale the panel's findings to the whole population? Did you implement a machine learning based tool?

- l580: I am not sure what the matching procedure attempts to do. My best guess is that you are trying to make causal inferences about the effect of gender and other covariates on whether a PR will be merged or not. Can you please add a few lines about what you where trying to achieve with it? Also, what are the types of features used (factors, integers etc) esp for the "file type" feature? I understand that much of what I am discussing here is mostly presented in l306 onwards, but it helps to have the discussion about propensity score matching in one place, either here (which I recommend) or before.

Validity of the findings

I have one major issue with the paper in its current form: the fact that the observed differences are attributed to bias.

Bias is an overloaded word: in science, it usually means deviation from the truth, while in every day life it is a synonym for discrimination. There are various documented forms of bias that are related to research, both in a statistical sense (e.g. detection bias) and also relating to assumptions and pre-occupations on the researcher's part (various forms of cognitive bias). The authors acknowledge most (all?) of those in their threats to validity section. For bias (as in discrimination) to occur one group or individual needs to consciously or unconsciously ACT against another. Otherwise put, there needs to be a causal link between the fact that an individual is a woman whose profile details are open and the fact that their PRs will be rejected.

The authors do identify an *association* between women with open profile details and drop in PRs acceptance rates, but, in my opinion, they do not identify a causal link. They attempt twice to do so:

* With their propensity analysis, they attempt to control for covariates in the PR process, but in the end the gender profiles from both men and women have exactly the same acceptance rate (Fig 11). The authors may consider as bias the drop in acceptance rates between gender-neutral and gendered profiles for women (and corresponding increase in men); however, the drop is very low (3%) and possibly within the confidence intervals (Fig 11) of both groups.

* In the discussion, the authors attempt a step-wise elimination of possible reasons for the observed difference. Rejecting a set of theories that fail to explain a result does not automatically make a newly proposed one valid.

What I think the authors are measuring are differences in how the two genders perform on various activities wrt pull request handling, including their under/over-representation in the process. This is, in my opinion, an important finding on its own. But I do not believe this constitutes bias, at least given the evidence presented by the authors.

Given the use of GitHub, I would also avoid to extrapolate my findings to OSS in general.

Finally, one thing I am missing from the paper is an outlook: given the differences that the authors find, what would they propose projects to do?

---

## Round 0.2 · accepted · Accept

This paper is exemplary in many ways, including in its thoughtful handling of the feedback and concerns from the reviewers.

This paper is clearly ready for acceptance. Congratulations with this paper, and thank you for publishing it with PeerJ.

One of the reviewers has a suggestion for page 9, which you may want to consider for the final version.

Typo on line 371: "an documented" -> "a documented"

·

Basic reporting

The authors have strengthened the framing of the paper, the literature review, and relevance of the key research question. In doing so, they have clarified the contribution of the paper and highlight the value of a multidisciplinary research perspective. Combining social science with computer science research methods is a significant contribution to both fields.

One minor suggestion on page 9 - add a reference to the likeability penalty for women who exhibit the same leadership behaviors as men - https://www.ncbi.nlm.nih.gov/pubmed/17227153

Experimental design

no further comment.

Validity of the findings

Robust data methods, and controls and an appropriately nuanced interpretation of the results.

Additional comments

I hope there is widespread dissemination of these findings in computing, in social science, and in the OS community and that it will generate further discussion of the value of multidisciplinary approaches to addressing the under-representation of women in computing. Please let me know when it comes out and we at the Clayman institute for Gender research would love to spread the word about it.

Reviewer 4 ·

Basic reporting

I have reviewed the paper before and commented positively on how nicely it reads. The new version makes an already nice paper nicer to read.

Experimental design

No comment

Validity of the findings

I have gone through the author's extensive rebuttal to both my comments and the comments of other reviewers. I am mostly convinced by the authors responses/changes.

What I am still not convinced about is the fact that the authors attribute the differences in PR acceptance to bias. However, on absence of a better explanation or evidence to contradict the author's theory, I am willing to accept this interpretation. Focused replications, and perhaps a large scale experiment, are needed to strengthen the findings reported here.